# Salt-Induced Modulation of Ion Transport and PSII Photoprotection Determine the Salinity Tolerance of Amphidiploid Brassicas

**DOI:** 10.3390/plants12142590

**Published:** 2023-07-08

**Authors:** Nisma Farooq, Muhammad Omar Khan, Muhammad Zaheer Ahmed, Samia Fatima, Muhammad Asif Nawaz, Zainul Abideen, Brent L. Nielsen, Niaz Ahmad

**Affiliations:** 1National Institute for Biotechnology and Genetic Engineering College (NIBGE-C), Pakistan Institute of Engineering and Applied Sciences (PIEAS), Faisalabad 38000, Pakistan; nismaansari@yahoo.com (N.F.); m.omar.khan@live.com (M.O.K.); samiafatima313@gmail.com (S.F.); m.a_nawax@outlook.com (M.A.N.); 2Dr Muhammad Ajmal Khan Institute of Sustainable Halophyte Utilization, University of Karachi, Karachi 75270, Pakistan; mzahmed@uok.edu.pk (M.Z.A.); zuabideen@uok.edu.pk (Z.A.); 3Microbiology & Molecular Biology, Brigham Young University, Provo, UT 84602, USA

**Keywords:** salt tolerance, marginal lands, photosynthesis, photosystem II, *Brassica juncea*, *Brassica napus*, relative gene expression

## Abstract

*Brassica* species show varying levels of resistance to salt stress. To understand the genetics underlying these differential stress tolerance patterns in Brassicas, we exposed two widely cultivated amphidiploid *Brassica* species having different genomes, *Brassica juncea* (AABB, *n* = 18) and *Brassica napus* (AACC, *n* = 19), to elevated levels of NaCl concentration (300 mM, half the salinity of seawater). *B. juncea* produced more biomass, an increased chlorophyll content, and fewer accumulated sodium (Na^+^) and chloride (Cl^−^) ions in its photosynthesizing tissues. Chlorophyll fluorescence assays revealed that the reaction centers of PSII of *B. juncea* were more photoprotected and hence more active than those of *B. napus* under NaCl stress, which, in turn, resulted in a better PSII quantum efficiency, better utilization of photochemical energy with significantly reduced energy loss, and higher electron transport rates, even under stressful conditions. The expression of key genes responsible for salt tolerance (*NHX1* and *AVP1*, which are nuclear-encoded) and photosynthesis (*psbA*, *psaA*, *petB*, and *rbcL*, which are chloroplast-encoded) were monitored for their genetic differences underlying stress tolerance. Under NaCl stress, the expression of *NHX1*, *D1*, and *Rubisco* increased several folds in *B. juncea* plants compared to *B. napus*, highlighting differences in genetics between these two Brassicas. The higher photosynthetic potential under stress suggests that *B. juncea* is a promising candidate for genetic modifications and its cultivation on marginal lands.

## 1. Introduction

Salinity is one of the significant detrimental factors reducing crop cultivation, affecting over one billion hectares of the world′s farmland [1]. The accumulation of high salt concentrations in different plant parts severely limits crop yields, especially in arid and semi-arid areas. Excessive ion uptake, sodium (Na^+^), and chloride (Cl^−^), along with a reduction in water, induce cytotoxicity and nutritional imbalance, which affect plant growth and development [2]. The uptake of toxic salt ions affects plant growth in two back-to-back phases. The first “osmotic phase” decreases plant growth through a declined rate of soil water potential, while the second “ionic phase” arises as the old leaves begin to die when salt ions exceed the vacuole capacity and start to accumulate in the cytoplasm [3]. The retention of salt ions in the roots, serving to limit their transfer towards the upper parts, is also a mechanism for plant survival under salt stress [4]. Another stress survival approach is the reduction in biomass production through the downregulation of carbon assimilation [5,6]. In this case, plants allocate large amounts of assimilated carbon to the production of osmoprotectants instead of utilizing them for development [7]. The use and transportation of soluble sugars can help plants to eliminate the adverse effects of salinity and regulate photosynthesis through compatible solutes or ROS regulators via feedback inhibition [8].

Photosynthesis rates have been shown to correlate with the degree of salt stress. Photosystem II (PSII) is highly stress-sensitive. Upon exposure to salt stress, PSII efficiency decreases, and the amount of energy loss increases [9]. In the early stages of exposure to salt stress, plants close their stomata to adjust their transpiration rate to a minimum and water usage efficiency (WUE) to a maximum [10]. Crucial biochemical reactions such as carbon fixation, RuBP regeneration, and gaseous exchange inhibitions have been observed following long-term exposure to salinity [11,12]. In grasses, for example, biochemical limitation and stomatal closure seem to affect photosynthesis and inhibit CO_2_ assimilation by damaging the photosynthetic machinery [13,14]. To reduce salt-induced damage, there are many defensive mechanisms, including excess heat removal via the activation of the xanthophyll cycle, avoiding the over-reduction of linear electron transport (LET) by diverting cyclic electron flow (CEF) around PSI [12,15]. These biochemical and physiological responses of plants under salt stress are also linked with their underlying genetic mechanisms. Different groups of genes are involved in salt tolerance in plants, such as genes that regulate ion transporters (e.g., Na^+^/K^+^ and H^+^ transporters) and are involved in the biosynthesis of important components of photosystems [16]. The expression analysis of different genes allows for the classification of genotypes based on their capacity to control various components of the salt tolerance mechanism [17].

The global demand for food is growing daily with the increasing world population, while the area under crop cultivation is decreasing due to salinization and intensive urbanization. Worldwide, food production needs to rise by approximately 57% by 2050 to overcome the increasing gap between food demand and supply [18,19]. A pragmatic approach to this problem is the development of stress-tolerant plant species that can grow on marginal lands. This means that the future of agriculture would depend on such versions of field crops that could foster saline agriculture. *Brassica* species belong to the family *Brassicaceae* and are widely grown as oilseeds, vegetables, poultry and livestock feed, as well as condiments [20]. The seed is characterized by its substantial oil content, reaching up to 45%, making it highly suitable for consumption. It is widely regarded as a healthy food choice due to its abundance of unsaturated fatty acids, comprising up to 93% of its composition. Additionally, it contains relatively low levels of saturated fatty acids, further contributing to its nutritional value [21].

Most of the *Brassica* crop varieties show moderate salinity tolerance levels (7–8 dsm^−1^) [22]. Furthermore, amphidiploid species in the genus *Brassica*, i.e., *B. carinata*, *B. juncea*, and *B. napus*, have been reported to show higher salt resistance than the diploid progenitors such as *B. oleracea*, *B. nigra*, and *B. rapa* due to some undefined reasons [2,23]. As inter-species salt tolerance is ascribed to complex mechanisms, very little is known about it. However, this information is critical for selecting a commercially known variety for further genetic transformation experiments, particularly for developing stress-resilient plants. 

The purpose of this study was to investigate the photosynthetic performance of, and genetic variations between, two commonly cultivated *Brassica* species with distinct genomes, *B. juncea* (AABB) and *B. napus* (AACC), through exposure to high-salt-stress conditions. Our data show that *B. juncea* showed higher salt tolerance compared to *B. napus* and that the high level of salt tolerance displayed by *B. juncea* was due to its highly protective PSII reaction centers, which resulted in higher PSII efficiency and better energy utilization with reduced energy loss.

## 2. Results

### 2.1. Effect of Salinity on the Growth Parameters of Brassica

*Brassica* plants were grown in pots and were exposed to 300 mM NaCl to determine the effect of NaCl stress on plant growth by determining different growth parameters (Figure 1). Both species showed a significant decrease in plant height under NaCl stress. However, the reduction was much higher in the *B. napus* plants (−34% of control) compared to *B. juncea* (−12% of control; Figure 1a). The sizes of the leaves also showed marked decreases for both species; however, the reduction was significantly higher in the *B. napus* plants (Figure 1b). Likewise, a significant decrease in plant biomass was observed, with a much higher reduction in the *B. napus* plants (Figure 1c,d). The decline in the moisture content of the plants grown with 300 mM NaCl was also determined; however, no significant differences in this parameter were observed.

### 2.2. Effects of Salinity on Pigments and Ion Concentration

Plants of *B. napus* and *B. juncea* were grown in pots and exposed to 300 mM NaCl stress. The concentrations of photosynthetic pigments such as chlorophyll and carotenoids and the chlorophyll a to chlorophyll b ratio were determined. Under control conditions, all these parameters were higher in *B. napus* than in *B. juncea* (Figure 2) yet found to be statistically insignificant. However, when the plants were exposed to 300 mM NaCl, both species showed contrasting results. For example, salt stress resulted in significantly marked decreases in the levels of these pigments in *B. napus*. In *B. juncea*, the levels of these pigments did not fall below the control level but rather increased after the salt treatment. For instance, the chlorophyll contents were decreased to almost half in the *B. napus* plants while they doubled in *B. juncea* under NaCl stress (Figure 2a). In addition to light harvesting, carotenoids play many important roles in plants, including photoprotection and the biosynthesis of antioxidants. High NaCl treatment resulted in a considerable reduction in carotenoid contents in *B. napus* (−87%), indicating less photoprotection of the PSII reaction centers from oxidative stress. In contrast, the concentration of carotenoids under NaCl significantly increased in *B. juncea* (+67%; Figure 2b). The chlorophyll a/b ratio—a general indicator of the extent of stress that are plants subjected to—was slightly increased in *B. juncea* (+12%) after exposing the plants to NaCl stress, while a significant drop (−61%) was observed in the *B. napus* plants, an indication of severe oxidative stress.

The concentration of different ions (Na^+^, K^+^, and Cl^−^) was determined in the leaves of the plants of both species after NaCl treatment (Figure 3). The level of Na^+^ in the control leaf tissues was similar to that in both *B. napus* and *B. juncea*, but when the plants were treated with NaCl, the concentration of Na^+^ was significantly increased in both species but to different extents. For example, a 2-fold increase in the Na^+^ level was observed in *B. juncea*, whereas a 3-fold increase was noted in the *B. napus* plants. Both the *B. juncea* and *B. napus* plants accumulated almost equal concentrations of Cl^−^ ions under control conditions. A slight increase was recorded in the uptake of Cl^−^ in *B. juncea* leaves when exposed to NaCl stress. However, the concentration of chloride ions in the leaf tissue was sharply (+211% of control) increased in the *B. napus* plants after salt treatment. Both species also varied in their retention/depletion of K^+^ in their leaves after salt treatment. Under control conditions, the level of K^+^ was comparable in both *B. juncea* and *B. napus*. Salt treatment resulted in a considerably significant decline in the K^+^ level in the leaves of *B. napus* (−66%) as compared to that of the *B. juncea* plant leaves (−32%).

### 2.3. Effect of Salinity on PSII Photochemistry

Chlorophyll fluorescence analysis is a well-established method used to estimate functional changes in photosynthesis [24]. It provides detailed information about PSII efficiency in intact leaves under different environmental conditions [25]. Chlorophyll fluorescence was probed to evaluate the efficiency and fluorescence quenching of PSII, including both the photochemical and non-photochemical forms, in *B. napus* and *B. juncea* with and without salt stress (Figure 4). The maximum quantum efficiency of PSII photochemistry, F_v_/F_m_, was comparable in the two species under non-stress conditions. However, exposure to 300 mM NaCl significantly reduced PSII quantum efficiency in *B. napus* (−66%), indicating the salt-induced photoinhibition of PSII due to stress. In contrast, the salt treatment did not significantly affect the F_v_/F_m_ levels in *B. juncea* (Figure 4a). Due to photoinhibition, a reduction in the rate of electron transportation (ETR) was observed in both Brassicas. However, *B. napus* showed a much higher reduction in this parameter (−70% of control) than *B. juncea*, which showed only a mild reduction in ETR, confirming the stress-induced photoinhibition of PSII in *B. napus* (Figure 4b). Likewise, the effective quantum yield of PSII, as denoted by φPSII, was higher in *B. juncea* under control conditions. Exposure to NaCl stress reduced the PSII quantum yields in both species; however, high NaCl exposure caused substantial PSII quantum yield losses in *B. napus* (−75%). In contrast, the PSII quantum yield in *B. juncea* did not fall below the control level, even after 300 mM NaCl stress; rather, an increase in this parameter was observed after the salt stress (Figure 4c). 

When plants undergo stress, they protect themselves from stress by increasing the dissipation of absorbed energy as heat, termed as the non-photochemical quenching of chlorophyll fluorescence, measured as regulated energy dissipation (NPQ) and nor-regulated energy dissipation (Y(NO)) [26]. The NPQ was almost equal in the unstressed plants of both species. However, it was considerably increased in *B. napus*, consistent with the earlier observations of PSII photoinhibition (Figure 5a). In contrast, *B. juncea* did not show any increase in NPQ, but it decreased after salt stress, in line with the earlier observations of the higher photoprotection of PSII in *B. juncea*. In the unstressed plants, Y(NO) was higher in *B. juncea* than in *B. napus* and showed no significant increase after salt treatment. The *B. napus* plants showed a sharp increase in non-regulated heat dissipation (+121% of control; *p* ≤ 0.01%) when exposed to a 300 mM salt concentration (Figure 5b).

### 2.4. Effect of Salt Stress on Light Response Curves

Plants adapt themselves to environmental changes to sustain their growth and yield. These adaptation strategies may include changes in metabolism and photosynthesis [27,28]. Light response curves (LRCs) describe the impact of light variations on photosynthesis by providing information about the maximum photosynthetic capacity and yield [29]. Therefore, LRCs show how well plants acclimate their photosynthetic performance in variable environments when subjected to changes like salt stress [30]. 

The LRCs showed significant differences between the control and salt-stressed plants of *B. juncea* and *B. napus* (Figure 6). When the light intensity was increased, the ETR was also increased in both genotypes. However, this increase in the ETR was much higher in *B. juncea* than in *B. napus*. Contrary to the ETR, a decreasing trend was observed in both species for an effective quantum yield of PSII, φPSII. The means of Y(II) at the 0 mM NaCl concentration were similar in the two genotypes, whereas the mean was ~3-fold lower in the *B. napus* plants at 300 mM NaCl compared to *B. juncea*.

### 2.5. Effect of Salinity on the Relative Gene Expression of Different Genes

Since *B. juncea* and *B. napus* showed variable growth and photosynthetic performance when exposed to 300 mM NaCl, *B. juncea* appeared to be more salt-tolerant than *B. napus* in terms of growth and photosynthesis under stress. To explain the genetic differences in these two species, we examined the expression of different genes located in the nuclear genome (*NHX1* and *AVP1*) as well as chloroplast genome (*psbA*, *psaA*, *petB*, *rbcL*) involved in salt tolerance and photosynthesis. NHX1 and AVP1 are nuclear-encoded enzymes and responsible for the transportation of Na^+^/K^+^ and H^+^, respectively. *NHX1* encodes for the vacuolar Na^+^/H^+^ antiporter involved in ionic homeostasis, salt tolerance, and leaf development [31], while *AVP1* encodes for a vacuolar H^+^-pyrophosphatase, which enables plants to withstand drought and salt stress regimes [32]. The relative quantitative expression of the nuclear-encoding *NHX1* and *AVP1* genes was increased in both *B. juncea* and *B. napus* under high salt stress compared to the control. However, the transcript level of these genes was much higher in *B. juncea* than in *B. napus*. For example, the expression of *NHX1* was increased ~3-fold in *B. juncea* under stress while it was only increased 1.3-fold in *B. napus* (Figure 7). The *AVP1* gene showed a slight increase in *B. napus*, while a ~100% increase was observed in the *B. juncea* plants.

As it was observed that the PSII in *B. juncea* was found to be less affected by NaCl stress compared to that in *B. napus*, we were interested in determining the effects of salt stress on different photosynthetic complexes, such as PSI, PSII, the cytochrome b_6_f complex and RuBisCo (Ribulose-1,5-bisphosphate carboxylase/oxygenase), the key enzyme of the Kelvin cycle. For this, we studied the expression of chloroplast-encoded genes—*psbA*, *psaA*, *petB*, and *rbcL*—coding for the proteins which make the core components of these complexes (Figure 8). When the plants were exposed to salt stress, the expression of the *psbA* gene, which encodes D1, is the core component of the PSII reaction center, and is highly sensitive to stress [33], was increased in both species. However, this increase was twice as high in *B. juncea* than in *B. napus* (Figure 8a). Likewise, the expression of the PSI gene, *psaA*, was doubled in the *B. juncea* plants under salt stress while decreasing in *B. napus* (Figure 8b). The expression of the *petB* gene, which encodes a component of the Cyt b_6_f complex, was downregulated in *B. juncea* under NaCl stress while showing a marked increase in *B. napus* (Figure 8c). A 4-fold increase in the expression of the Kelvin cycle enzyme, RuBisCo, was observed in *B. juncea* under stress, while a slight increase was recorded in *B. napus* (Figure 8d).

## 3. Discussion

*Brassica* is an important source of edible oil and, like other crops, experiences yield reduction due to abiotic stresses, including salinity. Most *Brassica* crops are predominantly cultivated in arid and semi-arid areas and often experience saline stress [34]. In this study, we assessed the salt tolerance potential of two commonly cultivated *Brassica* species with distinct genomes, *B. juncea* and *B. napus*. Our data show that *B. juncea* is more salt-tolerant than *B. napus* due to its regulated ion transport and better PSII-protective mechanisms. A high degree of salt tolerance and better regeneration potential of *Brassica juncea* [21] make this genotype a suitable candidate for genetic manipulations aimed towards the development of climate-smart, stress-tolerant oilseed crops.

The exposure of plants to 300 mM NaCl significantly reduced the growth of both species. However, this growth was more depressed in *B. napus* than in *B. juncea* (Figure 1). Similar results for *B. napus* were reported in a study by Wu et al. [35], in which a decrease in the shoot and root lengths of *B. napus* at 200 mM NaCl was observed. Similar findings have been reported [36,37,38]. Rezaei et al. [39] stated that salinity stress also decreased the average leaf area in *Brassica*, a result which is similar to our findings. We observed a decrease in average leaf size in both species, but this reduction was much higher in the *B. napus* plants. The higher salt tolerance observed in *B. juncea* has long been recognized [40]. This differential salt tolerance among Brassicas seems to be controlled by QTLs [41]. 

A plant’s salt sensitivity or resilience reflects its internal ionic balance [42]. In Brassicas, salt tolerance correlates with the transcription of the Na^+^/H^+^ antiporter (*NHX1*) that transports Na^+^ into vacuoles from the cytosol [23,43]. Na^+^ and Cl^−^, particularly when accumulated in photosynthesizing tissues, have been found to be toxic to plants [44,45,46]. The accumulation of higher levels of Na^+^ and Cl^−^ in *B. napus* with a lower expression of *NHX1* and H-PPase (*AVP1*) as compared to *B. juncea* is indicative of ionic toxicity in the cytoplasm due to the insufficient vacuolar compartmentalization of toxic ions. 

The observed effect of salinity stress on photosynthetic pigments in *B. napus*, namely, depletion, could be a combination of the inhibition of chlorophyll biosynthesis and/or increased degradation under salinity. A reduction in chlorophyll pigments is a sign of oxidative stress [47]. It represents the photoprotective mechanism of salt-sensitive plants, through which they decrease their light absorbance via the depletion of their chlorophyll contents [48]. The maintenance of photosynthetic pigments in *B. juncea* reflects its high tolerance to NaCl stress, an observation in line with earlier studies [49,50]. As reported earlier, the elevated levels of chlorophyll could be due to the more significant number of chloroplasts in stressed leaves [51]. Carotenoids are pigments that act as antioxidants and protect membrane lipids from oxidative stress [5]. We observed an increase in the carotenoid contents in *B. juncea* and a decrease in *B. napus*, with carotenoids having a significant role in the photoprotective mechanism. Studies have reported increased carotenoid contents under salt stress in salt-tolerant cultivars of *B. juncea* [52] and other *Brassica* species [53]. 

In this work, chlorophyll fluorescence was employed to investigate the impact of salt stress on PSII efficiency in *Brassica* plants. Various PSII parameters that provide information on the performance of the photosynthetic apparatus were studied [54]. An increase in non-photochemical quenching parameters and a decrease in overall PSII performance and photochemical quenching parameters reflect a generalized response of plants to salt stress [55]. This generalized response was also documented in this work. For example, F_v_/F_m_, which we considered as the primary salt stress indicator [56,57], remained unaffected in *B. juncea* under stress but significantly decreased in *B. napus*. These findings are in line with results reported in different studies, where no change was found in this parameter after salt stress in *Thellungiella halophila* [58], salt-tolerant barley [59], and rice plants [60]. Moreover, lowered values of F_v_/F_m_ indicate the inactivation of PSII reaction centers due to stress [61,62,63]. As observed in this study, a decrease in F_v_/F_m_ in different *Brassica* species was also reported by Pavlovic et al. (2019) [23]. This implied that the *B. napus* PSII experienced more damage than that of *B. juncea*. This line of evidence is further supported by the increases in non-photochemical quenching parameters in *B. napus* under stress (Figure 4). These observations are in line with various studies that reported similar trends in chlorophyll fluorescence quenching parameters in different species [55,64,65]. The drop in ETR in *B. napus* is in line with earlier results, with different studies reporting a salt-stress-induced decrease in ETR and PSII efficiency [38,58,66]. Overall, the maintenance of higher F_v_/F_m_, ETR, and PSII efficiency in *B. juncea* highlights its ability to sustain a decent photosynthetic performance, even under high salt stress. Furthermore, the increases in PAR did not increase the ETR or PSII efficiency in *B. napus* (Figure 6). The light response curves (LRCs) were used to elucidate the response of photosynthesis to varying light intensities, dependent upon the nature of the species and environmental conditions, such as abiotic stress [67]. Similar results were observed in two different plant species of the family *Hymenophyllaceae* in a study that reported a decrease in PSII efficiency and an increase in NPQ with increasing intensities of light, a finding which is consistent with our results [68].

The differential expression of different genes points towards the different genetics in these two species. Studies have shown that the overexpression of both *NHX1* [16,17,69] and *AVP1* [69,70,71] confers tolerance against salt stress, including other abiotic stresses. The *psaA* and *psbA* genes encode for the chlorophyll apoproteins of PSI (A1) and PSII (D1), respectively. High production of A1 and D1 proteins in *B. juncea* might have contributed to better photosynthetic performance and salt tolerance in this species compared to *B. napus*. It was recently shown that the overexpression of D1 in Arabidopsis, rice, and tobacco made transgenic plants stress-tolerant, producing high yields under stress conditions [33,72]. An increased expression of *rbcL* indicates better CO_2_ fixation and good photosynthetic activity in *B. juncea* plants, even under high salinity [73]. These results are supported by the findings of Li et al. (2021), where *rbcL* expression was more significant under high light and salinity [74]. Likewise, the reduction in *PetB* in *B. juncea* was also in line with the findings of Wang et al., who reported that the expression of photosynthetic genes related to electron transport, such as *PetA*, *PetB*, and *PetE*, was significantly reduced under saline stress [75].

The performance of *B. juncea* in terms of growth, as well as the maintenance of photosynthetic pigments and ionic homeostasis, was far better than that of *B. napus* at the 300 mM NaCl concentration. Similarly, the high levels of F_v_/F_m_, PSII efficiency, and ETR, as well as the reduction in energy loss, as measured using NPQ and Y(NO), indicate that *B. juncea* is highly salt-tolerant, which makes it a good candidate for cultivation on marginal lands. Furthermore, the differential expression of genes under salt stress in *B. juncea* and *B. napus* implies genetic differences between these two species. The high salt tolerance of the commercial cultivar of *B. juncea* cv. Aari canola that was used in this study and its better regeneration potential, as reported earlier [21], would allow for the direct transformation of commercial cultivars other than Westar, which is used as a model for *Brassica*. The direct transformation of commercial cultivars offers several advantages over the traditional approaches for engineering a trait into the model and then transferring this trait into commercial cultivars through crossing and backcrossing.

## 4. Materials and Methods

### 4.1. Plant Growth Conditions

Seeds of *B. juncea* and *B. napus* were obtained from the Oilseed Research Institute, Faisalabad, and grown under controlled temperature, light, and humidity (relative humidity: 32 ± 5%; temperature: 25/18 °C day/night; 12 h day/light; light intensity: 500 ± 50 μmol photons m^−2^ s^−1^). The seeds were germinated in plastic pots (25 cm height and 19 cm diameter) filled with loamy soil and irrigated with 0.5X Hoagland solution.

### 4.2. Salinity Treatment, Experimental Design, and Growth Parameters

In total, 24 plants of each species were divided into two (02) groups. The plants were sub-irrigated with 2 L of 0.5X Hoagland solution of pH 6.5–7.0 with or without 300 mM NaCl in 12 cm × 30 cm trays [76]. The salinity treatment was started when the plants were around 6 cm in height with five (05) leaves, using 50 mM NaCl day^−1^ increments to avoid osmotic shock. The salt concentrations were measured daily with a refractometer (ATAGO S-10E, Tokyo, Japan). The water loss was adjusted by replenishing it to the required level with distilled water. 

The plants were harvested after exposure to the given concentration of NaCl after two (02) weeks. We measured different plant growth parameters such as fresh weight and plant and root lengths. The fresh weight was recorded immediately after harvest. The plant samples were dried in an oven at 80 °C for 48 h for the dry weight measurement. The following equation was used to calculate the water content (WC) in the leaf and root tissues:WC (%) = [(FW − DW)/FW] × 100

The leaf area was determined using ImageJ [77]. The experiments were carried out in three replicates, and all measurements were carried out in three biological replicates.

### 4.3. Determination of Ion (Na^+^, K^+^, Cl^−^) Accumulation

The concentration of different ions (Na^+^, K^+^, Cl^−^) in the leaves was determined using the hot water extraction method [78]. Briefly, leaf samples were dried and powdered, mixing a fraction (0.1 g) with 10 mL water followed by homogenization at 80 °C for 4 h. The resulting mixture was filtered, followed by dilution with distilled water for the measurement of the ion concentration. The concentration of cations (Na^+^ and K^+^) was measured using a flame photometer (Intech model I-66, Wallonia, Belgium), while the measurement of Cl^−^ ions was carried out using an ion-selective electrode and BANTE instrument.

### 4.4. Leaf Pigments

The chlorophyll content in the fourth and fifth leaves of the plant was estimated using SPAD 502 (Konica Minolta, Osaka, Japan). The concentration of different photosynthetic leaf pigments such as chlorophyll a (Chl_a_), chlorophyll b (Chl_b_), total chlorophyll (Chl_t_), and carotenoids (C_x+c_) was determined by placing leaf discs of equal size and weight into 4 mL ethanol under dark conditions. The absorbance was read at 470, 644, and 662 nm [79]. The concentrations were calculated using the following formulae:Chl_a_ = 9.784 D662 − 0.99 D644
Chl_b_ = 21.42 D644 − 4.65 D662
C_x+c_ = (1000 D470 − 63.14 Chl_b_)/214

### 4.5. Chlorophyll Fluorescence Measurements

Chlorophyll fluorescence measurements were conducted using a PAM 2500 instrument (Walz). The plants were dark-adapted for 20—30 min to allow for the dissipation of electrons from the photosystems. Minimal fluorescence (F_o_) and maximal fluorescence (F_m_) were measured using actinic light (<0.1 µmol photons m^−2^ s^−1^) and a saturating pulse of 10,000 µmol photons m^−2^ s^−1^ for 0.6 s, respectively [59]. The maximum quantum efficiency of PSII (F_v_/F_m_) was calculated using F_v_/F_m_ = (F_m_ − F_o_)/F_m_.

The steady-state (F_s_) and maximal fluorescence (F_m_′) were measured in light-adapted plants. Different parameters such as the effective photochemical quantum yield of PSII (Y(II)), photochemical fluorescence quenching (qP), non-photochemical fluorescence quenching (NPQ), the quantum yield of regulated heat dissipation (Y(NPQ), the quantum yield of non-regulated heat dissipation (Y(NO)), and the relative electron transport rate (rETR) were calculated using the following formulae [80,81,82,83]:Y(II) = (F_m_′ − F_s_)/F_m_′
qP = F_m_′ − F/F_m_′ − F_o_′ 
NPQ = F_m_′/F_m_′^−1^

Y(NPQ) = (F/F_m_′) − (F/F_m_)
Y(NO) = F/F_m_

The values of Y(NPQ) and Y(NO) were calculated at a PPFD ≈ 700 µmol photons m^−2^ s^−1^ [84]:rETR = φPSII × PPFD × 0.5 × 0.84(1)
where φPSII is the effective quantum yield of PSII, PPFD is the photosynthetic photon flux density incident on the leaf surface, 0.5 represents the fraction of light absorbed by the PSII assuming an equal light energy distribution between PSII and PSI, and 0.84 represents the average proportion of light absorbed by the leaf [80].

### 4.6. RNA Extraction and qRT-PCR Analysis

Total RNA was extracted from the leaves of the treated (300 mM) and non–treated (0 mM) *B. juncea* and *B. napus* plants using the Tri-reagent method. For this purpose, approximately 100 mg of fresh leaf sample was ground in 1 mL of Tri-reagent using a pre-chilled mortar and pestle and kept on ice until further processing. The samples were incubated at room temperature for 5 min and then centrifuged at 13,000 rpm for 10 min at 4 °C. The supernatant was collected in a fresh microfuge tube, and 200 μL chloroform was added to it, followed by vigorous shaking for 15 s, and it was then kept at room temperature for 5–10 min. This was followed by centrifugation again at 13,000 rpm for 15 min at 4 °C. The clear aqueous phase was carefully transferred into a new tube, and 250 μL chilled isopropanol and 250 μL chilled 0.8 M sodium citrate solutions were added to the tubes, followed by inversion 6–8 times for gentle mixing. The samples were then incubated at room temperature for 5–10 min and centrifuged at 13,000 rpm for 10 min at 4 °C. The RNA pellet was obtained and washed with 1 mL of 75% ethanol. The tubes were centrifuged at 10,000 rpm for 5 min at 4 °C. The ethanol was removed, and the pellets were kept for air drying for 5 min at room temperature. The RNA was dissolved in 50 μL of autoclaved nuclease-free water and stored at −80 °C for further use. The isolated RNA was treated with DNase-1 to remove any potential DNA contaminations, and sample purity was estimated by calculating the A260/280 ratios followed by visualization on an agarose gel. Approximately 5 μg of total RNA was used to synthesize the first strand of cDNA using the RevertAid™ Reverse Transcriptase kit (Fermentas, Heidelberg, Germany), following the manufacturer’s protocol. The synthesized cDNA was diluted for further use in qRT-PCR analysis.

The primers for different nuclear and chloroplast genes were synthesized commercially (Table 1). The qRT-PCR reaction was carried out in a Bio-Rad CFX96TM Touch Real-Time PCR detection system (Bio-Rad, Hercules, CA, USA) in a 15 μL reaction volume comprising 7.5 μL of BrightGreen 2X qPCR MasterMix (ABM, Vancouver, BC, Canada), 0.4 μL of each of forward and reverse primer, 2 μL of diluted cDNA, and 4.7 μL of sterile nuclease-free distilled H_2_O. The PCR conditions for all the genes were optimized as follows: initial denaturation at 95 °C for 5 min followed by 35 cycles of 95 °C for 30s, 55 °C for 30s, and 72 °C for 30s. Melt curve analysis was executed using constant heating from 55 °C to 95 °C followed by 95 °C for 5s. All reactions were carried out in two technical replicates. The Actin-β (Act-β) gene was used as an internal control. The relative expression of each gene was calculated using the 2^−∆∆Ct^ method or Livak method [85], which directly uses the threshold cycles (Ct) to measure the fold change in gene expression, as given below:∆CT_(test)_ = CT_(target, test)_ − CT_(ref, test)_
∆CT_(calibrator)_ = CT_(target, calibrator)_ − CT_(ref, calibrator)_
∆∆CT = ∆CT_(test)_ − ∆CT_(calibrator)_
2^−∆∆CT^ = Normalized expression ratio

### 4.7. Statistical Analyses

All statistical analyses were carried out in an R environment 4.1.3 for Windows. The data on the obtained traits were analyzed for significance through either one-way or two-way ANOVA to analyze differences between groups and variables. The R package, Agricolae, was used to carry out a pairwise comparison of means using Tukey’s honest significant difference (HSD) post hoc test.

## 5. Conclusions

In conclusion, this study illustrated that *Brassica juncea* exhibits greater salt tolerance than *Brassica napus*. Our data provide strong evidence for this, citing regulated ion transport, effective photosystem II (PSII) protective mechanisms, and robust genetic attributes as pivotal factors. Notably, *B. juncea* demonstrated superior capabilities in maintaining ionic homeostasis, retaining photosynthetic pigments, and expressing genes vital for salt stress resilience compared to *B. napus* when exposed to 300 mM NaCl stress. We have found that this higher tolerance can be attributed to the regulated compartmentalization of toxic ions through the enhanced expression of the *NHX1* and *AVP1* genes in *B. juncea*. Our research also highlighted the capacity of *B. juncea* to maintain an efficient photosynthetic performance under high salt stress, together with its ability to sustain higher F_v_/F_m_, ETR, and PSII efficiency levels. Furthermore, *B. juncea* demonstrated higher resistance to oxidative stress and a more effective photoprotective mechanism, evidenced by the preservation of photosynthetic pigments and the increase in carotenoid content, which significantly shielded the plants from oxidative damage. 

Our research highlights the potential of *B. juncea* as a climate-resilient crop that is suitable for cultivation in arid and semi-arid regions subject to high-salinity conditions. Further studies are recommended to explore the detailed genetic mechanisms behind these differences in salt tolerance between *B. juncea* and *B. napus*. 

## Figures and Tables

**Figure 1 plants-12-02590-f001:**
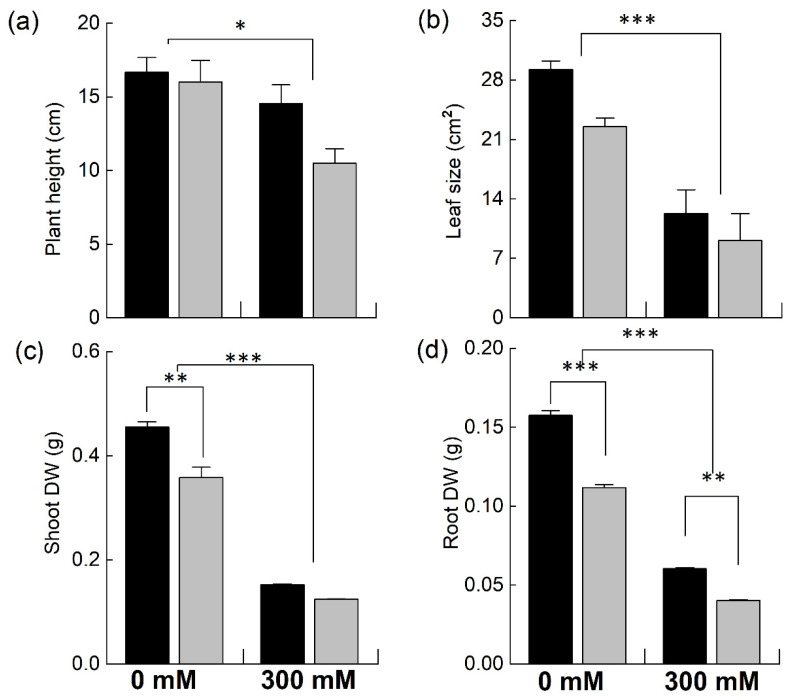
Comparison of different growth attributes of *Brassica* species under NaCl stress. Five-week-old plants of *Brassica juncea* (black bars) and *Brassica napus* (grey bars) were exposed to either 0 or 300 mM NaCl. Different growth parameters, such as plant height (**a**), leaf area (**b**), shoot dry weight (**c**), and root dry weight (**d**), were determined after 15 days of the treatment. Data points represent the means ± SD of three biological replicates. Significant differences are shown with different asterisks (***, **, * at *p* = 0.001, 0.01, 0.05, respectively).

**Figure 2 plants-12-02590-f002:**
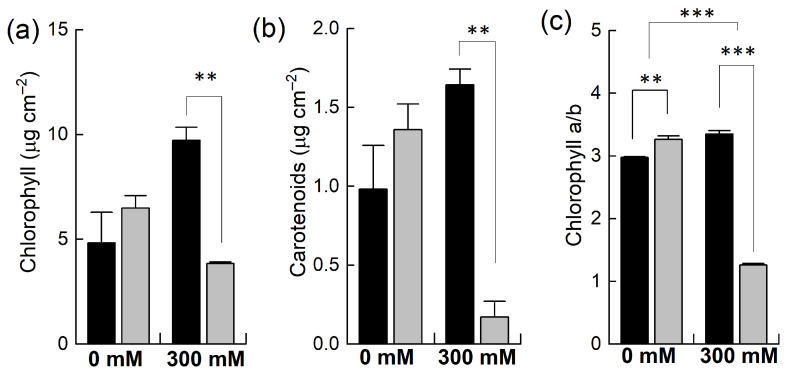
Effect of NaCl stress on different photosynthetic pigments. Plants of *Brassica juncea* (black bars) and *Brassica napus* (grey bars) were exposed to either 0 or 300 mM NaCl and concentrations of different photosynthetic pigments, including chlorophyll (**a**) and carotenoids (**b**), and the ratio of chlorophyll a/b (**c**) was measured after 15 days. Data points represent the means ± SD of three biological replicates. Significant differences are shown with different asterisks (***, ** at *p* = 0.001, 0.01, respectively).

**Figure 3 plants-12-02590-f003:**
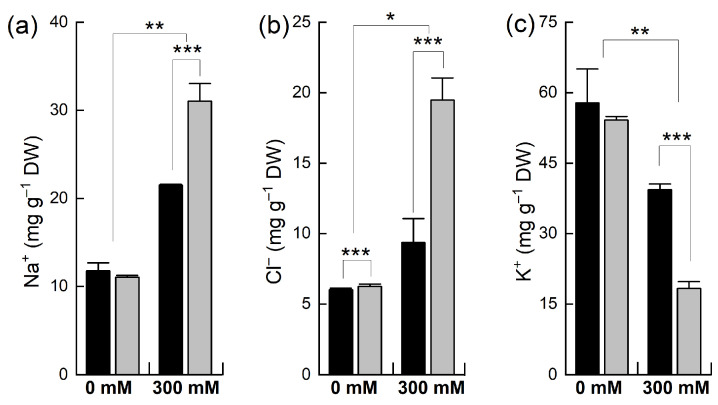
Accumulation of different ions in the leaves of *Brassica* plants under NaCl stress. Five-week-old plants of *Brassica juncea* (black bars) and *Brassica napus* (grey bars) were exposed to either 0 or 300 mM NaCl and concentrations of different ions, Na^+^ (**a**), Cl^−^ (**b**), and K^+^ (**c**), were measured after 15 days, as described in the methods. Data points represent the means ± SD of three biological replicates. Statistically significant differences are shown with different asterisks (***, **, * at *p* = 0.001, 0.01, 0.05, respectively).

**Figure 4 plants-12-02590-f004:**
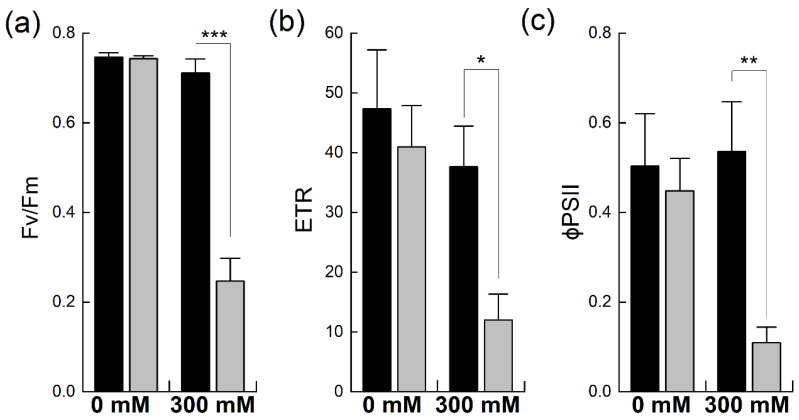
Effect of NaCl stress on PSII efficiency in *Brassica*. Plants of *Brassica juncea* (black bars) and *Brassica napus* (grey bars) were exposed to either 0 or 300 mM NaCl and different photosynthesis parameters such as the maximum quantum efficiency of PSII, F_v_/F_m_ (**a**), electron transport rate, ETR (**b**) and the quantum yield of PSII, φPSII (**c**) were measured. Data points represent the means ± SD of three biological replicates. Significant differences are shown with different asterisks (***, **, * at *p* = 0.001, 0.01, 0.05, respectively).

**Figure 5 plants-12-02590-f005:**
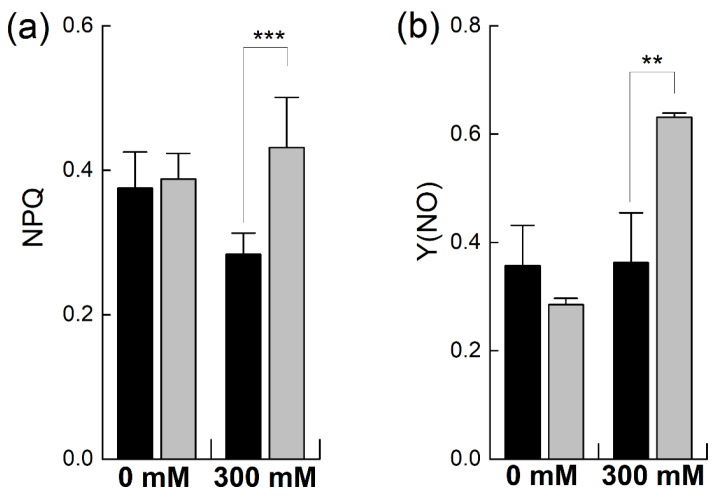
Effect of NaCl stress on non-photochemical PSII chlorophyll fluorescence in *Brassica.* Plants of *Brassica juncea* (black bars) and *Brassica napus* (grey bars) were exposed to either 0 or 300 mM NaCl and non-photochemical quenching of chlorophyll fluorescence was determined by recording regulated heat dissipation, NPQ (**a**), and nor-regulated energy dissipation, (Y(NO) (**b**). Data points represent the means ± SD of three biological replicates. Significant differences are shown with different asterisks (***, ** at *p* = 0.001 and 0.01, respectively).

**Figure 6 plants-12-02590-f006:**
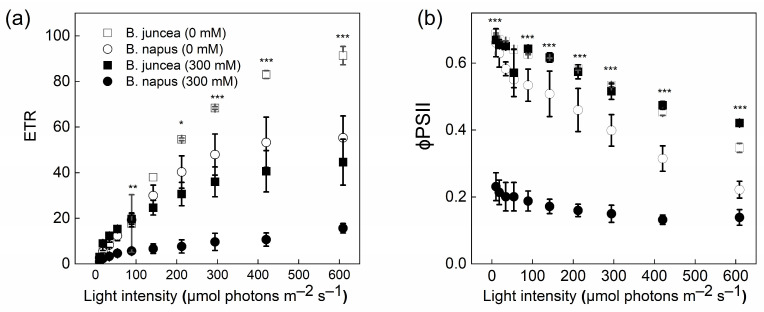
Electron transport rate and effective PSII efficiency in *Brassica* plants. *Brassica juncea* (black bars) and *Brassica napus* (grey bars) plants exposed to either 0 or 300 mM NaCl were used for the determination of the electron transport rate, ETR (**a**) and effective PSII efficiency, φPSII (**b**), in response to different light intensities at a CO_2_ partial pressure of 400 ppm. Data points represent the means ± SD of three biological replicates. Significant differences are shown with different asterisks (***, **, * at *p* = 0.001, 0.05 and 0.05, respectively).

**Figure 7 plants-12-02590-f007:**
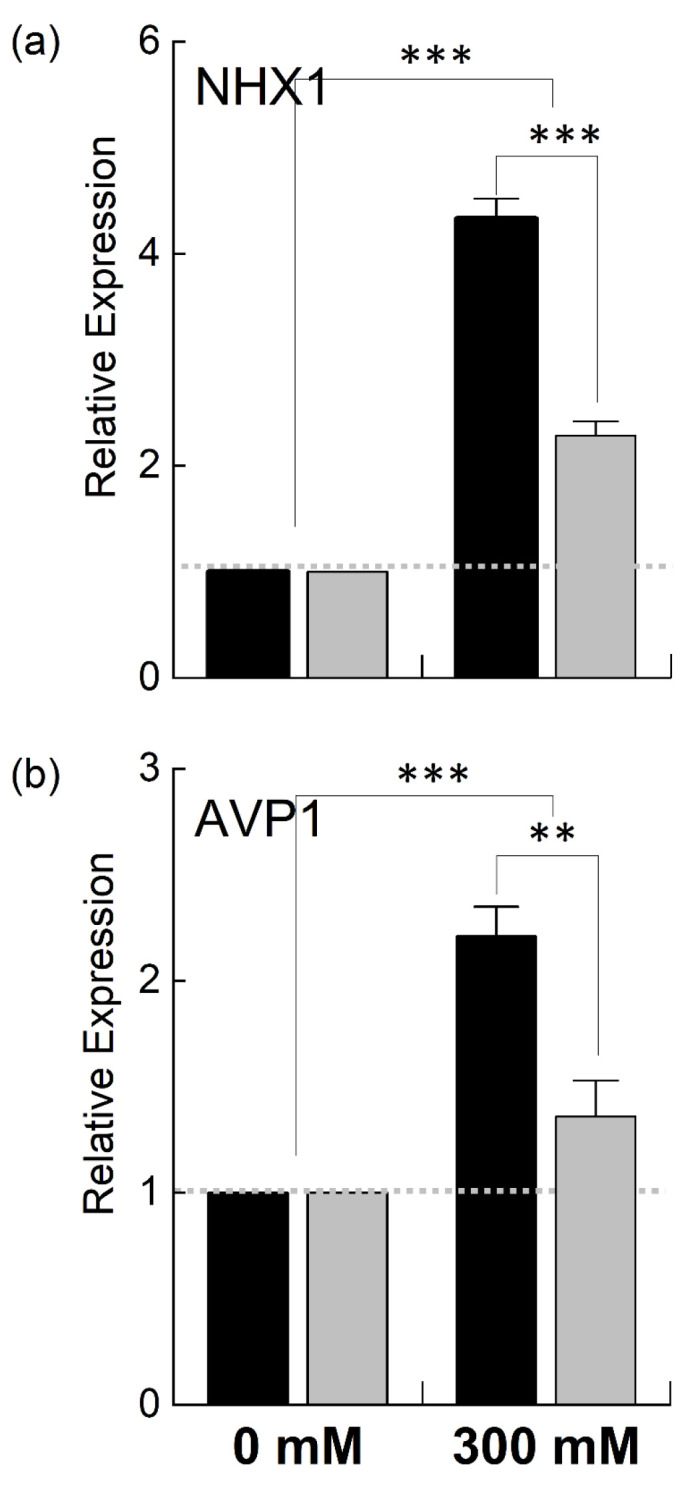
Relative expression of nuclear genes under salt stress. Fully expanded leaves of *Brassica juncea* (black bars) and *Brassica napus* (grey bars) plants exposed to either 0 or 300 mM NaCl were used for the determination of *NHX1* (**a**) and *AVP1* (**b**) expression levels. The dotted grey line represents the expression level of the genes in unstressed plants. Data points represent the means ± SD of three biological replicates. Statistically significant differences are shown with different asterisks (***, **, at *p* = 0.001, 0.01, respectively).

**Figure 8 plants-12-02590-f008:**
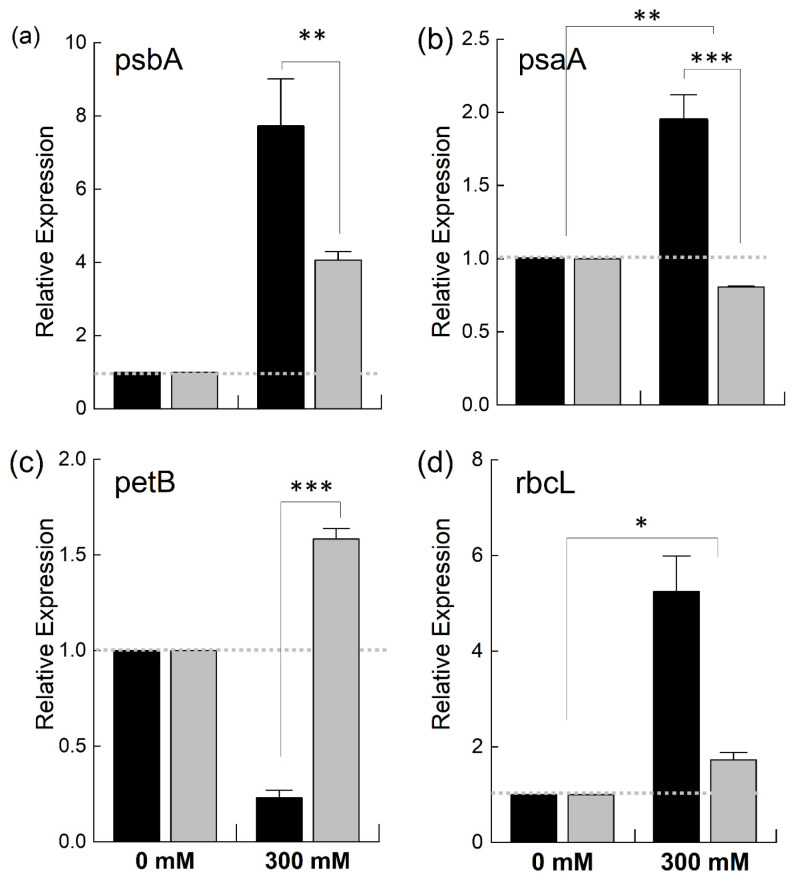
Relative expression of different chloroplast genes under salt stress. Fully expanded leaves of *Brassica juncea* (black bars) and *Brassica napus* (grey bars) plants exposed to either 0 or 300 mM NaCl were used to determine the expression levels of *psbA* (**a**), *psaA* (**b**), *petB* (**c**), and *rbcL* (**d**). The dotted grey lines represent the expression level of the genes in unstressed plants. Data points are the means ± SD of three biological replicates. Statistically significant differences are shown with different asterisks (***, **, * at *p* = 0.001, 0.01, 0.05, respectively).

**Table 1 plants-12-02590-t001:** List of primers used in this study for quantitative real-time PCR.

Sr. No.	Primer Name	Sequence (5′-3′)
1	AVP1-F	TCAGAGCCACACAAGGCAG
2	AVP1-R	GTGGCAAAGAAGGGAGCAAAG
3	NHX1-F	CAGTCTTGTATTCGGAGA
4	NHX1-R	AGCAGCTTCATGGTTAAGG
5	Act-β-F	TGTGACAATGGAACTGGAAT
6	Act-B-R	GACCCATCCCAACCATGA
7	psbA-F	AATTTTAGAGAGACGCGAAAGC
8	psbA-R	TCAAAACACCAAACCATCCA
9	psaA-F	AAGTTGCTCCTGCTACTCAGC
10	psaA-R	GCCCATGTTGTGGCAATTC
11	petB	GAGGCTTTTGCTTCTGTTCAA
12	PetB-R	GCAGGATCATCATTAGGACCA
13	rbcL-F	TGTTGGATTCAAAGCTGGTG
14	rbcL-R	TTGAGGAGTTACTCGGAATGC

## Data Availability

Not applicable.

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
