# Peer review of "Salt-Induced Modulation of Ion Transport and PSII Photoprotection Determine the Salinity Tolerance of Amphidiploid Brassicas"

_plants, 2023, doi:10.3390/plants12142590_

Round 1

Reviewer 1 Report

In this manuscript, the authors exposed two widely cultivated amphidiploid Brassica species having different genomes, Brassica juncea (AABB, n = 18) and Brassica napus (AACC, n = 19), to elevated levels of NaCl concentration (300 mM; half of the seawater salinity). They observed that B. juncea produced more biomass, increased chlorophyll content, and accumulated fewer sodium (Na+) and chloride (Cl–) ions in its photosynthesizing tissues perhaps because of a more active and photoprotected PSII compared to B. napus. Analyses of key genes associated with salt tolerance (NHX1 and AVP1) and photosynthesis (psbA, psaA, petB, rbcL) revealed that they were more responsive to salt stress in B. juncea plants compared to B. napus. In general, the research appeared well conducted, however some points need to be corrected or clarified.

Lines 75 and 75. Revise this sentence.

In Figure 1. Only plant height in B. juncea appeared less affected by salt stress in comparison to B. napus. The other data (leaf size; shoot DW; root DW) are not clear for me. Please, clarify this point.

Lines 113 and 114. The authors state that “Under control conditions, all these parameters were higher in B. napus than in B. juncea (Figure 2).” However, the differences appears not significant statistically, thus, it is important to consider this observation.

Lines 139 and 140. In the sentence: “For example, a 2-fold increase in Na+ level was observed in B. juncea, whereas a 3-fold increase was noted in B. napus plants” I think that Na+ level increases were approximately 1-fold and 2 fold, respectively.

Figure 3 legend is correct? “Accumulation of different ions in the leaves and roots of Brassica plants under NaCl stress.” However, in the text, authors present data only from leaves…

Why no statistic was applied in data of the Figure 6?, If data were obtained from biological triplicates.

Line 223 and 224. The authors state that “we examined the expression of different nuclear and chloroplast genes involved in salt tolerance and photosynthesis.” If I understood, I think that authors want to say that “they examined the expression of different nuclear genes that encoded chloroplast and extra-chloroplast proteins involved in salt tolerance and photosynthesis”. Or they also evaluated genes from chloroplast genome?

I think it is important to include in results one sentence explaining the NHX and AVP function in this study. In addition, the NHX and AVP proteins are encoded by gene families constituted by several members in different plant species. Why the authors choice only NHX1 and AVP1 genes to study their expression?

I think that the authors could evaluate better the title of this manuscript since that other cellular components could also contribute critically in salinity tolerance and they were not evaluate here. Thus, affirms that “only salt-induced modulation of ion transport and PSII photoprotection determine the salinity tolerance among amphidiploid Brassicas” is not exactly correct. Perhaps, change “determine” by “corroborate critically with” could be an option.

Line 359; Why the day/night temperature was the same?

Line 374. Reference for ImageJ?

No information about RNA isolation was given. Have the authors tested the RNA integrity and contaminations with genomic DNA, or other compounds? 

With regard to qPCR reaction, why the authors used the same annealing temperature (55 °C) for all primer pairs? Have the authors performed gradient temperature to certify about the best annealing temperature of each primer pair?

The authors used a single reference gene (Actin-β) to normalize the data, however no information/reference was provided about the expression stability of this gene in the tested conditions of this manuscript. This point is critical to reliability of expression data.

Minor editing of English language required

Author Response

Comments and Suggestions for Authors

In this manuscript, the authors exposed two widely cultivated amphidiploid Brassica species having different genomes, Brassica juncea (AABB, n = 18) and Brassica napus (AACC, n = 19), to elevated levels of NaCl concentration (300 mM; half of the seawater salinity). They observed that B. juncea produced more biomass, increased chlorophyll content, and accumulated fewer sodium (Na+) and chloride (Cl–) ions in its photosynthesizing tissues perhaps because of a more active and photoprotected PSII compared to B. napus. Analyses of key genes associated with salt tolerance (NHX1 and AVP1) and photosynthesis (psbA, psaA, petB, rbcL) revealed that they were more responsive to salt stress in B. juncea plants compared to B. napus. In general, the research appeared well conducted, however some points need to be corrected or clarified.

Response: We are thankful to the reviewer for reading the manuscript critically and suggesting areas of improvement to enhance the clarity of our manuscript. We have taken into account all the changes suggested by the learned reviewer. Below is our point-by-point response to each comment.

Concern 1: Lines 75 and 75. Revise this sentence.

Response: The sentence has been revised (Line 76-80)

Concern 2: In Figure 1. Only plant height in B. juncea appeared less affected by salt stress in comparison to B. napus. The other data (leaf size; shoot DW; root DW) are not clear to me. Please, clarify this point.

Response: Figure 1 implies that all growth attributes were affected by salt stress in both of the species however, the effect of salt stress was more pronounced in B. napus than B. juncea. B. juncea tolerated increased NaCl levels better than B. napus in terms of growth attributes.

Concern 3: Lines 113 and 114. The authors state that “Under control conditions, all these parameters were higher in B. napus than in B. juncea (Figure 2).” However, the differences appears not significant statistically, thus, it is important to consider this observation.

Response: We have rephrased the sentences (Line 118-119)

Concern 4: Lines 139 and 140. In the sentence: “For example, a 2-fold increase in Na+ level was observed in B. juncea, whereas a 3-fold increase was noted in B. napus plants” I think that Na+ level increases were approximately 1-fold and 2 fold, respectively.

Response: We want to draw the attention of the reviewer that the fold change difference between Na uptake is 2x and 3x at 200 mM and 300 mM NaCl conc between B. juncea and B. napus plant leaves. It was calculated by dividing the values obtained at 200 mM and 300 mM by the values with their respective controls.

Consern 5: Figure 3 legend is correct? “Accumulation of different ions in the leaves and roots of Brassica plants under NaCl stress.” However, in the text, authors present data only from leaves…

Response: We have corrected the mistake, thanks for pointing out this mistake of ours.

Concern 6: Why no statistic was applied in data of the Figure 6?, If data were obtained from biological triplicates.

Response: The statistics were applied to these data as well. However, to avoid the clutter we did not show it on the figure. Anyways, we have now added statistical information and updated the legend as well. Kindly see the revised version of the draft/fig

Concern 7: Line 223 and 224. The authors state that “we examined the expression of different nuclear and chloroplast genes involved in salt tolerance and photosynthesis.” If I understood, I think that authors want to say that “they examined the expression of different nuclear genes that encoded chloroplast and extra-chloroplast proteins involved in salt tolerance and photosynthesis”. Or they also evaluated genes from chloroplast genome?

Response: We would like to affirm here that the genes from both genomes (nuclear genome: NHX1 and AVP1) and (chloroplast genome: psbA, psaA, petB, rbcL) were included in the analyses. We have made it clear in the manuscript as well (Please see lines 23, 25, 229—235, 252)

Concern 8: I think it is important to include in results one sentence explaining the NHX and AVP function in this study. In addition, the NHX and AVP proteins are encoded by gene families constituted by several members in different plant species. Why the authors choice only NHX1 and AVP1 genes to study their expression?

Response: Information related to the function of NHX1 and AVP1 has been incorporated (Please see lines 229-235). The rationale regarding the choice of NHX1 and AVP1 genes is their well-established role in salt tolerance in different species.

Concern 9: I think that the authors could evaluate better the title of this manuscript since that other cellular components could also contribute critically in salinity tolerance and they were not evaluate here. Thus, affirms that “only salt-induced modulation of ion transport and PSII photoprotection determine the salinity tolerance among amphidiploid Brassicas” is not exactly correct. Perhaps, change “determine” by “corroborate critically with” could be an option.

Response: We have considered changing the wording of the title but we feel that the current title is most accurate while the suggested wording is cumbersome and lengthy. The current title does not say ‘only’ in it and the work is thus open to other possible contributions. So we have left the title the same.

 Concern 10: Line 359; Why the day/night temperature was the same?

Response: The day/night temperature was not the same. It looks like a typing mistake. We have corrected it (Line 366). Thanks for pointing out the error to us.

Concern 11: Line 374. Reference for ImageJ?

Response: We have provided the said reference (Line 376)

 Concern 12: No information about RNA isolation was given. Have the authors tested the RNA integrity and contaminations with genomic DNA, or other compounds? 

Response: Although, it is a routine for RNA isolation. Keeping in view the reviewer's suggestion, we have included the missing information about RNA isolation, integrity and potential contamination (Line 417-438).

 Concern 13: With regard to qPCR reaction, why the authors used the same annealing temperature (55 °C) for all primer pairs? Have the authors performed gradient temperature to certify about the best annealing temperature of each primer pair?

Response: The primers have been meticulously designed in a way to ensure that their annealing temperatures are closely aligned with each other.

Concern 14: The authors used a single reference gene (Actin-β) to normalize the data, however no information/reference was provided about the expression stability of this gene in the tested conditions of this manuscript. This point is critical to reliability of expression data.

Response: We selected this gene because its expression stability is well established under different conditions (Acta physiologiae plantarum 2020 42:1-4; Frontiers in Plant Science 2021 12:683891)

Concern 15: Comments on the Quality of English Language (Minor editing of English language required)

Response: We checked the entire manuscript for the improvement of the English language, and made necessary updates.

Reviewer 2 Report

Dear Editors,

Thank you so much for choosing me as a reviewer of the manuscript. ID plants-2477474 entitled: ,,Salt-induced modulation of ion transport and PSII photoprotection determine the salinity tolerance among amphidiploid Brassicas’’.

I hope that my comments will help Authors to improve their manuscript.

Detailed comments to the manuscript.

The clear scientific hypothesis together with the answer for the questions tated as scientific hypothesis should be given.

Bibliography should be prepared strictly according to the guidelines for Authors. Reference list contains editorial mistakes. For example: Once the abbreviated titles of the journal are used, but the other time the full names are presented. Please go through the whole reference list and do needed change

Author Response

Dear Editors,

Thank you so much for choosing me as a reviewer of the manuscript. ID plants-2477474 entitled: ,,Salt-induced modulation of ion transport and PSII photoprotection determine the salinity tolerance among amphidiploid Brassicas’’.

I hope that my comments will help Authors to improve their manuscript.

Response: We are thankful to the reviewer for sparing his/her valuable time to read the manuscript thoroughly and suggesting improvements. We have incorporated all the suggestions he made, and it has helped us to improve the manuscript's quality. The point-by-point response to the reviewer’s comments is given below:

Concern 1: The clear scientific hypothesis together with the answer for the questions tated as scientific hypothesis should be given.

Response: Although the purpose of the study was already included, we have revised it to adjust the reviewer's suggestion (Lines 88-94)

Concern 2: Bibliography should be prepared strictly according to the guidelines for Authors. Reference list contains editorial mistakes. For example: Once the abbreviated titles of the journal are used, but the other time the full names are presented. Please go through the whole reference list and do needed change

Response: We have removed all the errors in the bibliography and updated it accordingly. Thanks for the suggestion

Reviewer 3 Report

This work makes a good impression and can be accepted for publication in Plants MDPI with minor revisions.

Lines 26-28 “The higher regeneration potential, as reported earlier…” This statement does not apply to this study and should be removed.

Lines 83-89. Here the authors report what has been achieved in their work. However, this should be moved to a discussion or conclusion. At the end of the Introduction, they usually write, for example, “the aim of the study was ....” And this needs to be more clearly and precisely reformulated.

As for the Conclusion, it is, in my opinion, too large. The last few sentences could have been removed.

Author Response

Comments and Suggestions for Authors

This work makes a good impression and can be accepted for publication in Plants MDPI with minor revisions.

Response: We are thankful to the learned reviewer for the feedback and comments for improving the quality of the manuscript. We have taken into account all the suggestions of the reviewer, and have revised the manuscript accordingly. Below is a point-by-point response to each of the reviewer’s suggestions:

Concern 1: Lines 26-28 “The higher regeneration potential, as reported earlier…” This statement does not apply to this study and should be removed.

Response: We have removed the said part from the sentence (Line 26-28)

Concern 2: Lines 83-89. Here the authors report what has been achieved in their work. However, this should be moved to a discussion or conclusion. At the end of the Introduction, they usually write, for example, “the aim of the study was ....” And this needs to be more clearly and precisely reformulated.

Response: By taking into account the reviewer's suggestion, we have rephrased this part (Line 88-94)

Concern 3: As for the Conclusion, it is, in my opinion, too large. The last few sentences could have been removed.

Response: We have shortened the conclusion section by removing the suggested sentences as suggested (Line 459-476)